

# Regionalising rainfall–runoff modelling for predicting daily runoff in continental Australia

Hongxia Li[1]and Yongqiang Zhang[2*]

[1]State Key Laboratory of Hydraulics and Mountain River Engineering, Sichuan University, Chengdu, 610065,
China;

[2]CSIRO Land and Water, PO BOX 1666, Canberra ACT 2601, Australia

[*]*Correspondence to*: Yongqiang Zhang (yongqiang.zhang@csiro.au)

**Abstract.** Numerous regionalisation studies have been conducted to predict the runoff time series in ungauged catchments. However, there are few studies investigating their benefits for predicting runoff time series on a continental scale. This study uses four regionalisation approaches, including spatial proximity (SP), gridded SP, integrated similarity (IS) and gridded IS, to regionalise two rainfall–runoff models (SIMHYD and Xinanjiang) for 605 unregulated catchments distributed across Australia. The SP and IS approaches are used for directly predicting catchment streamflow; the gridded SP and gridded IS approaches are used for predicting runoff at each $0.05° \times 0.05°$ grid cell for continental Australia, which is then aggregated for each catchment. The IS and gridded IS approaches use five properties to build similarity indices, including three physical properties (an aridity index, a fraction of forest ratio and the mean annual air temperature) and two rainfall indices (rainfall seasonality and the standard deviation of daily rainfall). The two rainfall–runoff models show consistent regionalisation results, and there is a marginal difference among the four regionalisation approaches in the wet and densely located catchments. However, the gridded IS approach outperforms the other three in the dry and sparsely located catchments, and it overcomes the unnatural tessellated effect obtained from the gridded SP approach. Use of the gridded IS approach together with rainfall–runoff modelling for predicting runoff on a continental scale is highly recommended. Extra predictors should be included to build similarity indices in other regions, such as the high latitude northern hemisphere or high elevation regions.



## 1. Introduction

Predicting runoff time series is the key of surface water hydrology. Traditionally, most continental runoff simulations are conducted using global hydrological models and global land surface models (Zhou et al., 2012; Beck et al., 2016a). However, their simulations are considerably poorer than the predictions from traditional

rainfall–runoff modelling (Zhang et al., 2016), especially in small catchments (Gudmundsson et al., 2012). It is typically difficult to calibrate land surface models/global hydrological models because these models are complex in their model structures, and water flux simulations for the land surface models are not their major aim. Some modellers seek to maintain model parameters with physical meaning and are reluctant to calibrate their global models (Beck et al., 2016a). This limits their runoff simulations and predictions. Compared to

these global models, rainfall–runoff modelling is simple in model structure and is easily calibrated and used for prediction in ungauged catchments (Blöschl and Sivapalan, 1995). However, using rainfall–runoff modelling to predict runoff at a continental or global scale has seldom been investigated over the last 20 years.

Rainfall–runoff models are widely used for runoff prediction in ungauged catchments using a regionalisation approach to transfer information from gauged (donor) to ungauged (target) catchments (Blöschl and Sivapalan,

1995). The most popular regionalisation approaches include (1) a spatial proximity approach (SP) (Reager and Famiglietti, 2009; Parajka et al., 2005b), in which the entire set of parameter values are transferred from the geographically closest catchment to the target ungauged catchment; (2) a physical similarity approach (PS) (Samuel et al., 2011; Reichl et al., 2009; Samaniego et al., 2010), in which the entire set of parameter values are transferred from a physically similar catchment whose attributes (climatic and physical) are similar to those of the target ungauged one; (3) a regression method (Reg) (Young, 2006), in which a relationship

between the parameters calibrated on gauged catchments and catchment descriptors or attributes is established and then the parameter values for the ungauged catchments are estimated from its attributes and the established relationship; (4) a regional calibration approach (RC), in which the models are calibrated simultaneously against observations in multiple catchments across a wide region to obtain a more generalisable parameter set

for all catchments (Zhang et al., 2011; Parajka et al., 2007); (5) the hydrological signature similarity approach (HSS), in which the parameters are transferred from a catchment that has a similar hydrograph shape, such as the runoff coefficient and baseflow index (Masih et al., 2010; Yadav et al., 2007; Wagener et al., 2007); and



(6) the integrated similarity (IS) methods, such as the integration of spatial proximity and physical similarity (Zhang and Chiew, 2009a), in which several regionalisation approaches are combined based on the effectiveness of the different methods.

**Table 1.** A summary of the regionalisation approaches conducted using large datasets. SP: spatial proximity

5  approach; PS: physical similarity approach; Reg: regression method; RC: regional calibration approach; HSS: hydrological signature similarity approach; and IS: integrated similarity

| Studies | Region | Catchment number | Model | Regionalisation method |
|---|---|---|---|---|
| Merz and Blöschl (2004) | Austria | 308 | HBV | Reg |
| McIntyre et al. (2005) | UK | 127 | PDM | PS and Reg |
| Parajka et al. (2005) | Austria | 320 | HBV | SP, PS, and Reg |
| Kay et al. (2006) | UK | 119 | PDM and TATE | PS and Reg |
| Young (2006) | UK | 260 | PDM | Reg and PS |
| Oudin et al. (2008) | French | 913 | GR4J and TOPMO | SP, PS, and Reg |
| Reager and Famiglietti (2009) | Southeast Australia | 210 | Xinanjiang | SP and PS |
| Reichl et al. (2009) | Southeast Australia | 184 | SIMHYD | SP, PS, and Reg |
| Zhang and Chiew (2009a) | Southeast Australia | 210 | Xinanjiang and SIMHYD | SP, PS, and IS |
| Li et al. (2010) | Southeast Australia | 227 | Three-parameter FDC | SP, PS, Reg, and HSS |
| Masih et al. (2010) | Karkheh river basin, Western part of Iran | 11 | HBV | HSS |
| Zhang et al. (2014a) | southeast Australia | 228 | An index model and GR4J | SP, Reg |
| Bock et al. (2016) | USA | 1575 | MWBM | RC |
| Viviroli and Seibert (2015) | Switzerland | 49 | PREVAH | HSS |
| Steinschneider et al. (2015) | USA | 73 | abcd | Combining Reg and SP |
| Patil and Stieglitz (2015) | USA | 756 | EXP-HYDRO | SP |
| Beck et al. (2016b) | Global | 1787 | HBV | HSS |
| Zhang et al. (2016) | Global | 664 | GR4J and SIMHYD | SP |
| This study | Australian continent | 605 | SIMHYD, Xinanjiang | SP, IS, gridded SP, and gridded IS |

All of these regionalisation methods have been applied in many catchments, including 308 catchments in

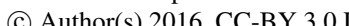


Austria (Merz and Blöschl, 2004), 260 catchments in the UK (Young, 2006), 913 catchments in France (Oudin et al., 2008), and 227 catchments in southeast Australia (Li et al., 2010). Table 1 summarises the regionalisation approaches conducted by using large datasets. Many attempts have been made to demonstrate the applicability of various regionalisation approaches (Table 1). However, each regionalisation approach does not always

perform consistently and may produce different conclusions between studies. This is because first, all of the previously mentioned regionalisation approaches have methodological limitations that preclude straightforward application (Beck et al., 2016b). Second, these studies used different hydrological models, different catchment sets, different catchment descriptors, and most studies conducted at a catchment or regional scale use a comparatively small number of catchments. Therefore, to fully understand the

performance of the various regionalisation approaches, it is necessary to have continental or global studies that cover more catchments.

Continental regionalisation studies have only started within the last one to two years. Bock et al. (2016) developed a parameter regionalisation scheme for a monthly water balance model by grouping the conterminous United States into 110 calibration regions based on similar parameter sensitivities and produced

parameter sets for each calibration region. Beck et al. (2016b) conducted the regionalisation of the Hydrologiska Byråns Vattenbalansavdelning (HBV) model parameters at the global scale on 1787 catchments using a hydrological signature similarity approach and found that the spatial patterns in regionalised parameter values corresponded well with spatial patterns in the climate. Zhang et al. (2016) evaluated two rainfall–runoff models (GR4J and SIMHYD) using a spatial proximity approach against a global dataset of streamflow from

644 catchments and ET from 98 flux towers. Although these studies were conducted on a larger scale using more catchments, they used only one regionalisation method, which may not fully demonstrate the effectiveness of regionalisation. Therefore, more efforts should be devoted to regionalisation methods in continental or global studies.

In this study, we used two rainfall–runoff models (SIMHYD and Xinanjiang) for predicting the daily runoff

time series for continental Australia. For each model, four regionalisation methods were used for the continental regionalisation study: two obtained from the SP approach and two obtained from the IS approach. The specific aims of this study include (1) to develop a gridded IS approach for runoff prediction in each grid



of continental Australia, (2) to compare the relative merits among the four regionalisation approaches, (3) to investigate if the relative merits are consistent between the two rainfall–runoff models, and (4) to stratify the regionalisation results using precipitation and regionalisation distance.

## 2. Data

Both models were driven by a daily meteorological time series of maximum temperature, minimum temperature, incoming solar radiation, actual vapour pressure and precipitation from 1975 to 2012 at $0.05° \times 0.05°$ (~ 5 km $\times$ 5 km) grid cells from the SILO Data Drill of the Queensland Department of Natural Resources and Water (www.nrw.gov.au/silo). The SILO data were interpolated from approximately 4600 point observations across Australia using the geostatistical methods described in Jeffrey et al. (2001). The ordinary

kriging method was used to interpolate daily and monthly precipitation, whereas the thin plate smoothing spline was used to interpolate other daily climate variables. Cross validation shows that the mean absolute error for maximum daily air temperature, minimum daily air temperature, vapour pressure, and precipitation at 1.0°C, 1.4°C, 0.15 kPa and 12.2 mm/month indicated reasonably good data quality (Jeffrey et al., 2001). The $0.05° \times 0.05°$ SILO spatial data were averaged across all of the grid cells within a catchment to produce

a catchment average time series for use in this study. The rainfall data are required as an input for the rainfall-runoff models. The other meteorological data were used to calculate potential evapotranspiration ($ET_p$) using the Priestley-Taylor model and to calculate actual evapotranspiration ($ET_a$) for the revised rainfall–runoff models.

The daily time series of streamflow data for the selected 605 unregulated catchments (50 to 5000 km$^2$) (Figure

1) were collated by Zhang et al. (2013). The streamflow data were quality assessed using quality codes and spike control methods. Data from 1975 to 2012 are used in this study.

The remote sensing leaf area index (LAI) data that were required to calculate $ET_a$ in the revised rainfall-runoff models were NOAA-AVHRR monthly leaf area index data at ~8 km resolution, which were obtained from Boston University.

The land cover data required to estimate aerodynamic conductance for the Penman-Monteith $ET_a$ equation were obtained from the MODIS land cover product and the yearly land cover classification product





(MOD12Q1) (http://edcdaac.usgs.gov/modis/mod12q1v4.asp). The dataset has 17 vegetation classes, which are defined according to the International Geosphere-Biosphere Programme.

The albedo data required to calculate net radiation were obtained from the 8 day MODIS MCD43B bidirectional reflectance distribution function product at 1 km resolution.

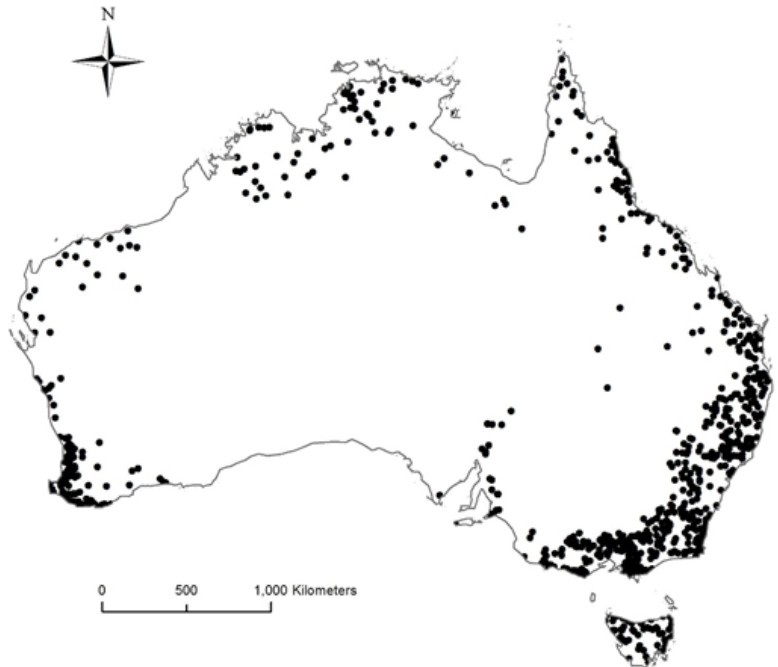

**Figure 1.** Geographic location of the 605 gauged catchments used in this study

All of the remote sensing and meteorological data were re-projected and resampled to obtain 0.05° gridded data. The gridded data in each catchment were then extracted and averaged to obtain an aggregate daily data

10    series for use in the modelling.

## 3. Model description and modelling experiments

This study uses the SIMHYD and Xinanjiang models. Their overall characteristics are described in Table 2. Their details are introduced in sections 3.1 and 3.2.





**Table 2.** An overview of the characteristics of the two hydrological models used in this study

| Characteristics | SIMHYD | Xinanjiang |
|---|---|---|
| Number of free parameters | 10 | 12 |
| Interception | A zero capacity interception store | No interception store |
| Evapotranspiration | PML evapotranspiration submodel | PML evapotranspiration submodel |
| Runoff production | A soil moisture accounting store; Infiltration excess surface runoff and saturation excess runoff | A soil moisture accounting store; Saturation excess runoff |
| Routing | A nonlinear routing store | Lag-and-route routing; A nonlinear routing store |
| Source | Zhang and Chiew (2009a) | Zhang and Chiew (2009a) |

### 3.1 SIMHYD and Xinanjiang models

Globally, the SIMHYD and Xinanjiang models are both widely used for rainfall–runoff modelling (Zhang and

Chiew, 2009b, a; Li et al., 2012; Li et al., 2013; Li et al., 2014; Viney et al., 2009b; Vaze et al., 2010; Chiew

et al., 2010; Zhou et al., 2013; Zhao, 1992). This study uses the revised versions of the SIMHYD and

Xinanjiang conceptual daily rainfall–runoff models (Zhang and Chiew, 2009b). The revised versions both

replace the original $ET_a$ sub-models using the Penman-Monteith-Leuning (PML) $ET_a$ model and can simulate

the runoff process response to precipitation, potential evapotranspiration and the leaf area index. The

Xinanjiang model has 12 parameters. The SIMHYD model has 10 parameters. The SIMHYD and Xinanjiang

models both perform better than their original versions for southeastern Australia (Zhang and Chiew, 2009b)

and therefore are used here for the prediction of continental runoff.

### 3.2. Model calibration

A global optimisation method, the genetic algorithm, was used to calibrate the model parameters for each of

the 605 unregulated catchments. The model calibration period is from 1975 to 2012 with the first two years

(i.e., 1975–1976) for spin up, and the prediction period from 1977 to 2012. The genetic algorithm used a

population size of 400 and the maximum generation of 100 for searching the optimum point, and the algorithm

normally converges to the optimum point at approximately 50 generations of searching.

The two models were calibrated to best reproduce the daily runoff by minimising the following objective



function (Viney et al., 2009b):

$$F_1 = (1 - NSE_{sqrt}) + 5 \left| \ln(1 + Bias) \right|^{2.5} , \tag{1}$$

where $NSE_{sqrt}$ is Nash–Sutcliffe Efficiency of the daily square-root-transformed runoff data, which

compromises weight when simulating high and low flows (Pena-Arancibia et al., 2015) and is expressed as:

$$NSE_{sqrt} = 1 - \frac{\sum_{i=1}^{M} \left( \sqrt{Q_{obs,i}} - \sqrt{Q_{sim,i}} \right)^2}{\sum_{i=1}^{M} \left( \sqrt{Q_{obs,i}} - \sqrt{\overline{Q_{obs}}} \right)^2} , \tag{2}$$

and *Bias* is

$$Bias = \frac{\sum_{i=1}^{M} Q_{sim,i} - \sum_{i=1}^{M} Q_{obs,i}}{\sum_{i=1}^{M} Q_{obs,i}} , \tag{3}$$

in which $Q_{sim}$ and $Q_{obs}$ are the simulated and observed daily runoff, respectively, $\overline{Q}_{obs}$ is the arithmetic mean

of the observed daily runoff, $i$ is the $i$th day, and $M$ is the total number of days sampled. $NSE_{sqrt}$ measures the

agreement between the modelled and observed daily values with $NSE_{sqrt} = 1.0$, which indicates perfect

agreement between all of the modelled and observed daily runoffs. The *Bias* measures the water balance error

between the modelled and observed mean annual runoff with *Bias* = 0, which indicates that the mean daily or

total runoff is the same as the observed runoff.

### 3.4. Regionalisation

To evaluate the two models for predicting daily runoff in 'ungauged' catchments, each of the 605 catchments

is left out and regarded as an 'ungauged' catchment, and the parameter sets obtained from the donor (or

contribution) catchments are used to simulate daily runoff in the targeted 'ungauged' catchment. Several

modelling studies show that the use of a donor number of five can achieve reasonable model performance for

the SIMHYD and Xinanjiang models and that model performance is approaching the optimum for the

catchments widely distributed in Europe, Australia, and the USA (Patil and Stieglitz, 2012; Oudin et al., 2008;

Viney et al., 2009c; Zhang and Chiew 2009a). Therefore, this study used five donors in the regionalisation for

predicting daily runoff, and the predictions shown hereafter are all using five donors.

Four regionalisation approaches are evaluated in this study including

(1)Spatial proximity aggregated in catchments (SP),





(2)Spatial proximity for each grid cell (gridded SP),

(3)Integrated similarity aggregated in catchments (IS), and

(4)Integrated similarity for each grid cell (gridded IS).

The SP approach uses the geographic distance ($D$) between the centroids of the target prediction catchment

and the donor catchment to select the five closest donors; the gridded SP approach predicts runoff in each grid

cell and then aggregates gridded runoff into catchment streamflow in which the five closest donors were

selected using the $D$ between the donor and the centroids of each targeted grid cell. The SP approach is a

commonly used regionalisation approach to predict runoff in ungauged catchments (Bardossy, 2007; McIntyre

et al., 2005; Merz and Bloschl, 2004; Parajka et al., 2005; Oudin et al., 2008).

For the IS and gridded IS approaches, we develop integrated similarity indices (SI) for determining the five

most similar donors for predicting catchment streamflow (the IS approach) or runoff from each grid (the

gridded IS approach), which can be expressed as

$$SI_j = \frac{(I_j - I_t)^2}{I_t},$$ (4)

where $SI_j$ is the similarity index for the donor catchment $j$, $I_t$ is the property value for the target ungauged

catchment (the IS approach) or grid cell (the gridded IS approach), and $I_j$ is the property value for the donor

catchment. This study uses five similarity properties, including three physical properties (the aridity index, the

fraction of forest ratio and the mean annual air temperature) and two rainfall indices (the rainfall seasonality

and the standard deviation of daily rainfall). These five properties are representative of climate and catchment

attributes and are also easily applicable to each grid cell.

The SP and IS approaches regionalise model parameters at the catchment scale and then predict catchment

streamflow. However, the gridded SP and gridded IS approaches regionalise model parameters and predict

runoff in each grid cell and then aggregate gridded runoff into catchment streamflow.

For each regionalisation approach, we used the same weighting method for the five closest (or most similar)

donors. The assumption is that the closer (or more similar) the donor to its target 'ungauged' catchment (grid

cell), the more weight given to this donor. The inverse distance weight approach is used for weighting each

donor using from the following equation:





$$w_j = \begin{cases} \dfrac{1}{D^p}, & \text{for SP and gridded SP approaches} \\[2mm] \dfrac{1}{SI^p}, & \text{for IS and gridded IS approaches} \end{cases}, \qquad (5)$$

where $p$ is a positive real number called the power parameter. This study used a value of 2 for $p$ (Zhang et al., 2014b).

For the SP and IS approaches, $Q_{sim}$ for each 'ungauged' catchment is obtained using

$$Q_{sim} = \frac{\sum_{j=1}^{5} Q_j \times w_j}{\sum_{j=1}^{5} w_j}, \qquad (6)$$

where $Q_j$ is the simulated streamflow using the parameter from the $j$ donor catchment.

For the gridded SP and gridded IS approaches, $Q_{sim}$ for each "ungauged" catchment is obtained using

$$Q_{sim} = \frac{\sum_{j=1}^{5} \left( \sum_{k=1}^{L} Q_k \times w_k \right) \times w_j}{\sum_{j=1}^{5} w_j}, \qquad (7)$$

where $Q_k$ is the simulated streamflow from the $k$ grid cell, $w_k$ is the proportion of the grid cell within the "ungauged" catchment and the sum of $w_k$ from the $L$ grid cells is 1.

## 3.5. Model evaluation

Three criteria are used for the model assessment of the regionalisation results: two $NSE$ metrics, including the $NSE$ of daily runoff and the $NSE_{sqrt}$ of daily square-root-transformed runoff described by Eq. (2), and the absolute model bias described by Eq. (3).

The model evaluation period is the same as the model calibration period, i.e., 1977–2012 for model regionalisation evaluation with 1975–1976 for model spin up.

## 3.6. Continental runoff prediction

The gridded SP and gridded IS approaches are applied to predict runoff in each $0.05° \times 0.05°$ grid cell. A total of 321, 457 grid cells of continental Australia are simulated for the period from 1975 to 2012.

## 4. Results

Figures 2–3 summarise the overall regionalisation approach performance in predicting daily runoff across all 605 catchments; Figure 4 summarises the regionalisation approach performance in predicting mean annual





runoff (i.e., model bias). Figures 5–6 show maps of the mean daily runoff in each $0.05° \times 0.05°$ grid cell simulated by the SIMHYD model (Figure 5) and the Xinanjiang model (Figure 6). Figures 7–11 show model predictions in different regimes, which are stratified using mean annual rainfall (Figures 7–8) and geographic distance (Figures 9–11).

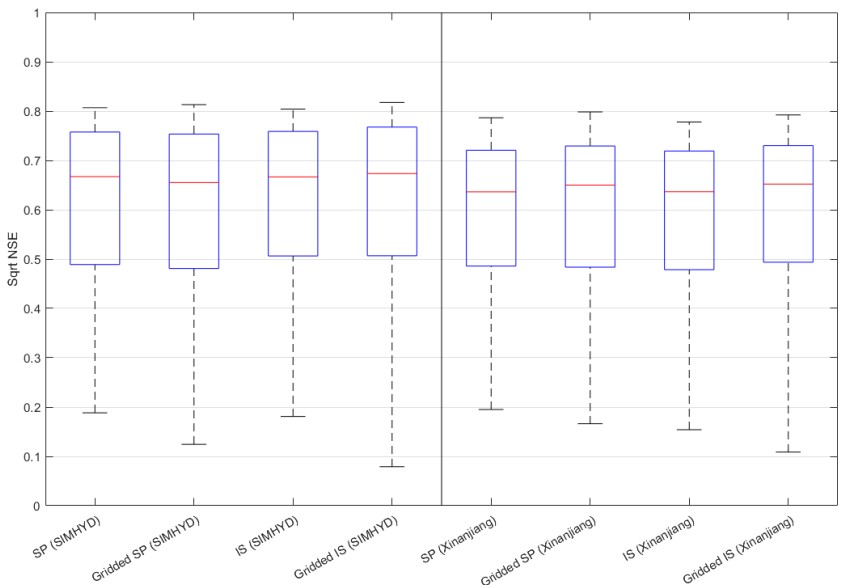

**Figure 2**. Summary of square root (sqrt) transferred Nash-Sutcliffe Efficiency (NSE$_{sqrt}$) values for the four regionalisation approaches. Whisker plots show the 10[th], 25[th], 50[th] (median), 75[th] and 90[th] results for the 605 catchments.

10    The four regionalisation methods show marginal differences in predicting the daily runoff in terms of NSE$_{sqrt}$ (Figure 2) and NSE (Figure 3). The median NSE$_{sqrt}$ for the SIMHYD model for the four methods (SP, gridded SP, IS, and gridded IS) is 0.67, 0.66, 0.67, and 0.67, respectively; the median NSE$_{sqrt}$ for the Xinanjiang model for the four methods is 0.64, 0.65, 0.64, and 0.65. The median NSE for the SIMHYD model for the four methods (SP, gridded SP, IS, and gridded IS) is 0.55, 0.56, 0.54, and 0.58, respectively; the median NSE for

15    the Xinanjiang model for the four methods is 0.54, 0.57, 0.54, and 0.57. This result suggests that the gridded SP approach is better than the SP approach, and the gridded IS approach slightly outperforms the IS approach. Overall, the gridded IS approach performs best.





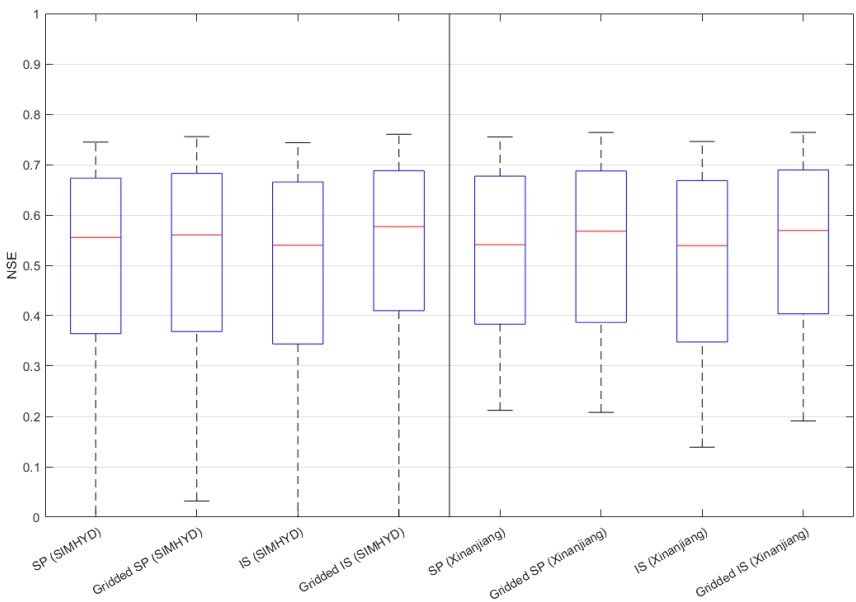

**Figure 3**. Summary of Nash-Sutcliffe Efficiency (NSE) values for the four regionalisation approaches.

5      Whisker plots show the 10[th], 25[th], 50[th] (median), 75[th] and 90[th] results for the 605 catchments.

Similarly, the four regionalisation methods show marginal differences in model bias (Figure 4). The median

absolute bias for the four methods (SP, gridded SP, IS, and gridded IS) is 0.20, 0.20, 0.23, 0.21 for the SIMHID

model and is 0.26, 0.230.30, and 0.23 for the Xinanjiang model, respectively.





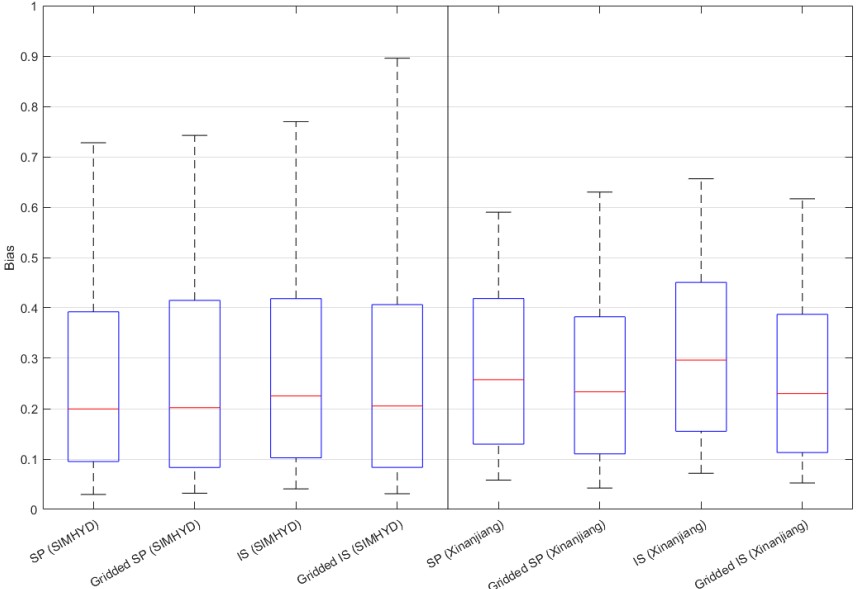

**Figure 4**. Summary of absolute bias values for the four regionalisation approaches. Whisker plots show the 10[th], 25[th], 50[th] (median), 75[th] and 90[th] results for the 605 catchments.

The SP and IS approaches can only be used at a catchment scale. In comparison, the two grid–based regionalisation approaches (gridded SP and gridded IS) can be used for predicting runoff not only at the catchment scale (results shown in Figures 2–4) but also in each grid cell. Figure 5 shows the mean daily runoff for 1977–2012 in each $0.05° \times 0.05°$ grid cell across continental Australia, which were obtained from the gridded SP approach (Figure 5A) and the gridded IS approach (Figure 5B) using the SIMHYD model. Overall,

the mean daily runoff obtained from the gridded SP approach shows similar spatial distribution to that of the gridded IS approach. However, the gridded SP approach shows the unnatural tessellated effect in inland Australia where the gauges are sparsely distributed. In comparison, the gridded IS approach exhibits natural variation in the mean daily runoff in the regions with sparse streamflow gauges. Figure 6 shows the same maps but using the Xinanjiang model. The results are consistent with those obtained from the SIMHYD model.

In the following paragraphs in this section, the gridded SP approach is further compared to the gridded IS approach in different regimes (Figures 7–11).



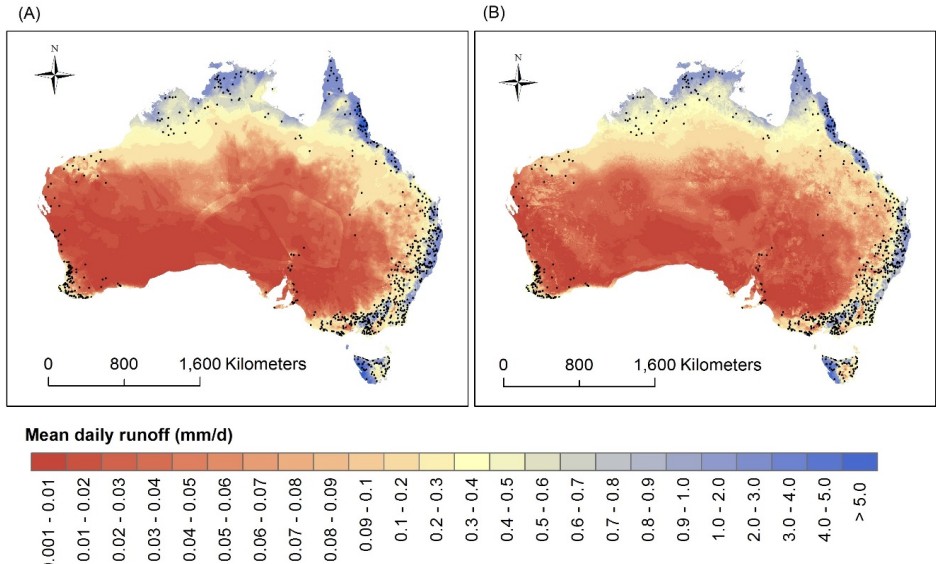

**Figure 5**. Spatial pattern of mean daily runoff at each $0.05° \times 0.05°$ grid cell simulated by spatial proximity (A) and integrated similarity (B) using the SIMHYD model. The 605 streamflow gauges are shown as dots on the two maps.

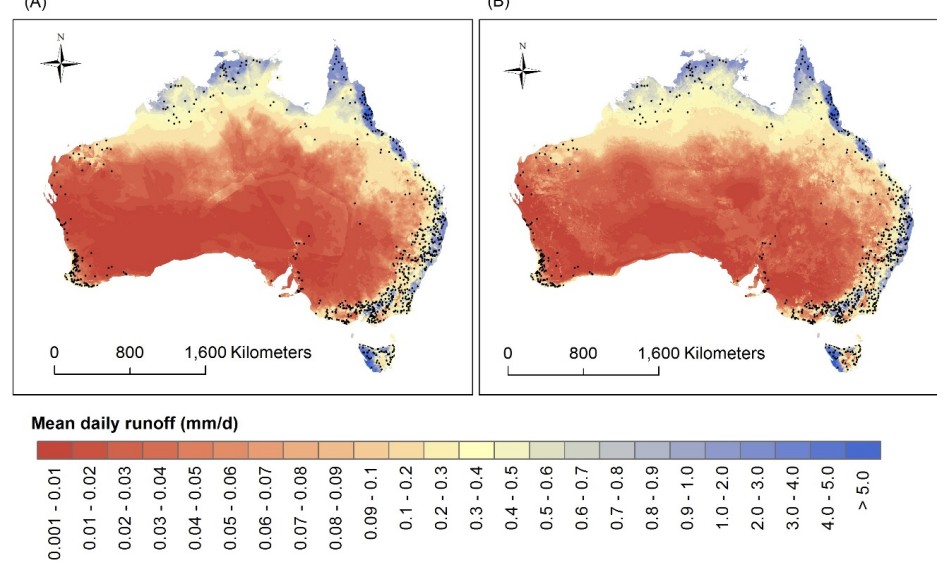

**Figure 6**. Spatial pattern of mean daily runoff at each $0.05° \times 0.05°$ grid cell simulated by spatial proximity (A) and integrated similarity (B) using the Xinanjiang model. The 605 streamflow gauges are shown as dots



in the two maps.

Figure 7 stratifies the NSE$_{sqrt}$ results into five groups using the mean annual rainfall ($P$) ($P < 600$ mm/yr, $600 \leq P < 800$ mm/yr, $800 \leq P < 1000$ mm/yr, $1000 \leq P < 1200$ mm/yr, and $P \geq 1200$ mm/yr). For both the

SIMHYD and Xinanjiang models, the IS and gridded IS approaches, respectively, outperform the SP and gridded SP approaches in the two dry catchment groups ($P < 600$ mm/yr and $600 \leq P < 800$ mm/yr), but are marginally different from the SP and gridded approaches in the wet catchment groups with a $P > 800$ mm/yr. Figure 8 shows bias stratification results using the same five P groups. Unlike NSE$_{sqrt}$, there is no clear bias difference between the IS and SP approaches and between the gridded IS and gridded SP approaches in each

climate group.

Figure 9 stratifies the NSE$_{sqrt}$ results into five groups using regionalisation distance ($D \leq 30$ km, $30 < D \leq 50$ km, $50 < D \leq 70$ km, $70 < D \leq 100$ km, and $D > 100$ km). For both the SIMHYD and Xinanjiang models, the IS and gridded IS approaches, respectively, outperform the SP and gridded SP approaches in the two groups with a large regionalisation distance ($70 < D \leq 100$ km and $D > 100$ km) but are marginally different from the

SP and gridded approaches in the other three groups. The same $D$ stratification is used to group the bias results, as shown in Figure 10. For both the SIMHYD and Xinanjiang models, the IS and gridded IS approaches, respectively, obtain less model bias than the SP and gridded SP approaches in the largest distance group (and $D > 100$ km) but are marginally different from the SP and gridded approaches in the other four groups. Figure 11 further compares the mean annual observed streamflow and the mean annual simulated streamflow in the

five regionalisation distance groups, which are obtained from the gridded SP (1st and 3rd rows) and gridded IS (2nd and 4th rows) approaches. In the group with D > 100 km, the Root Mean Squared Error between the observed and modelled (RMSE) obtained from the gridded IS approach is 23 mm/yr and 20 mm/yr less than that from the gridded SP approach for the SIMHYD and Xinanjiang model, respectively. The RMSE obtained from the two gridded approaches is marginally different in the other four groups.






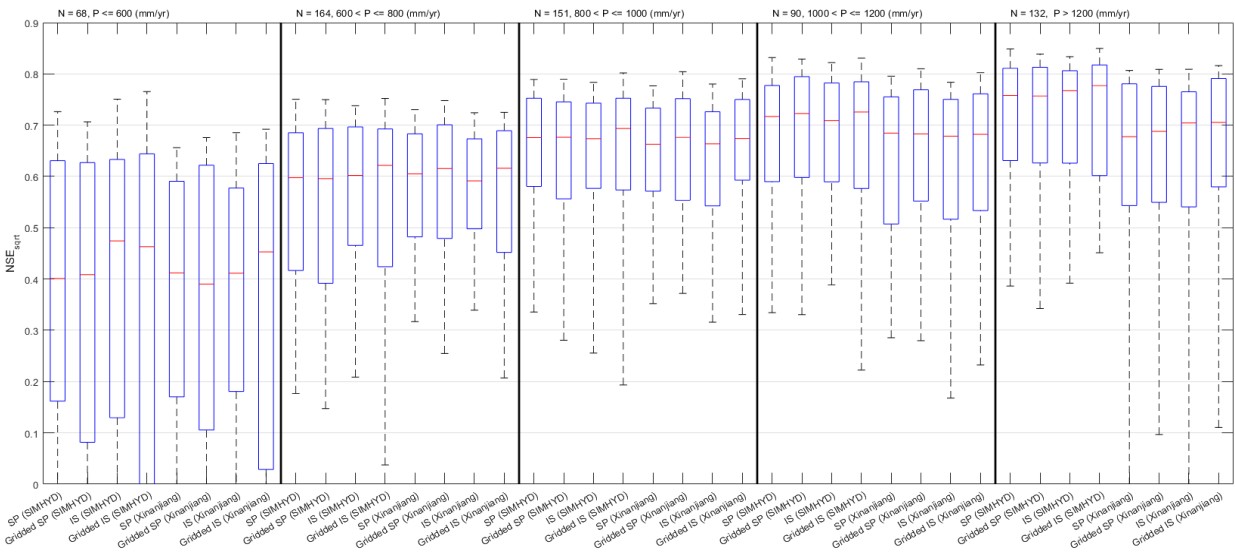

**Figure 7.** Stratification of NSE$_{srqt}$ regionalisation results using mean annual precipitation (mm/yr)

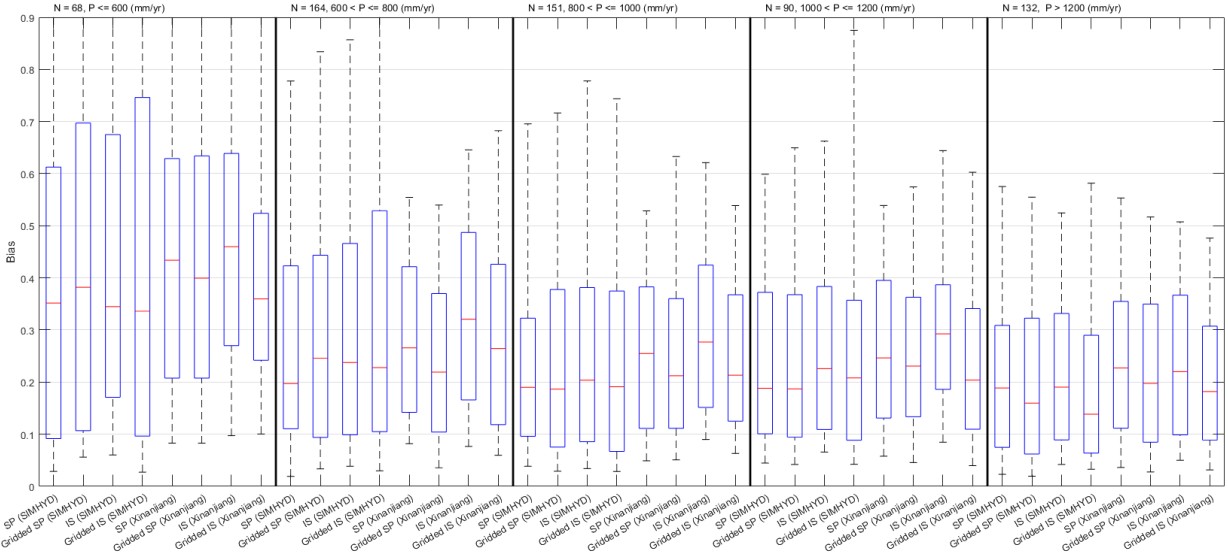

**Figure 8**. Stratification of bias regionalisation results using mean annual precipitation (mm/yr)





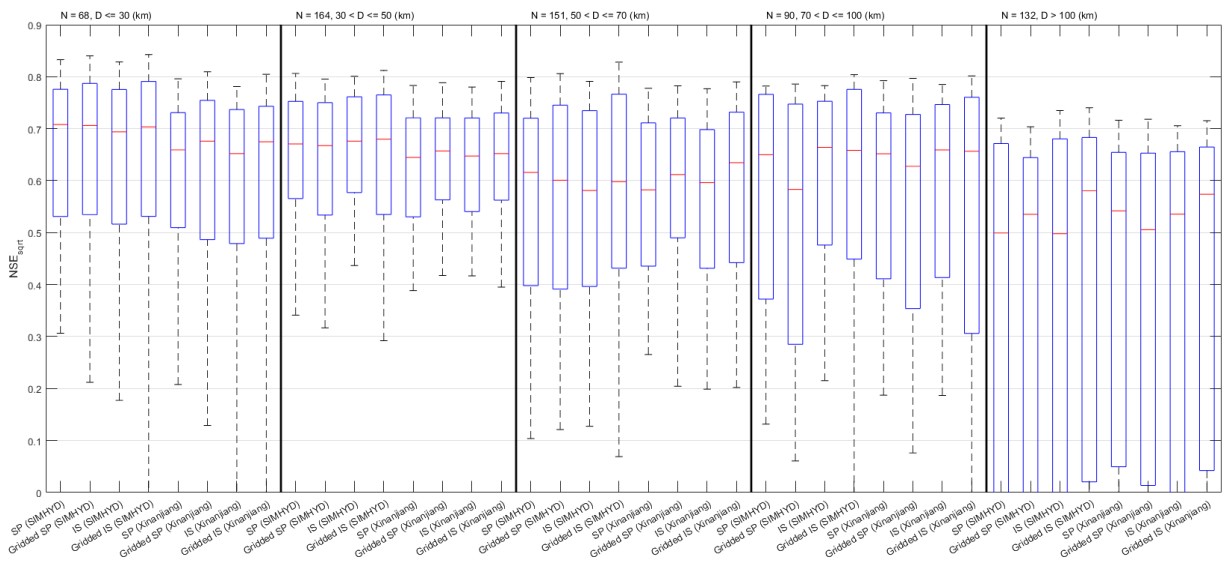

**Figure 9**. Stratification of NSE$_{srqt}$ regionalisation results using distance (km)

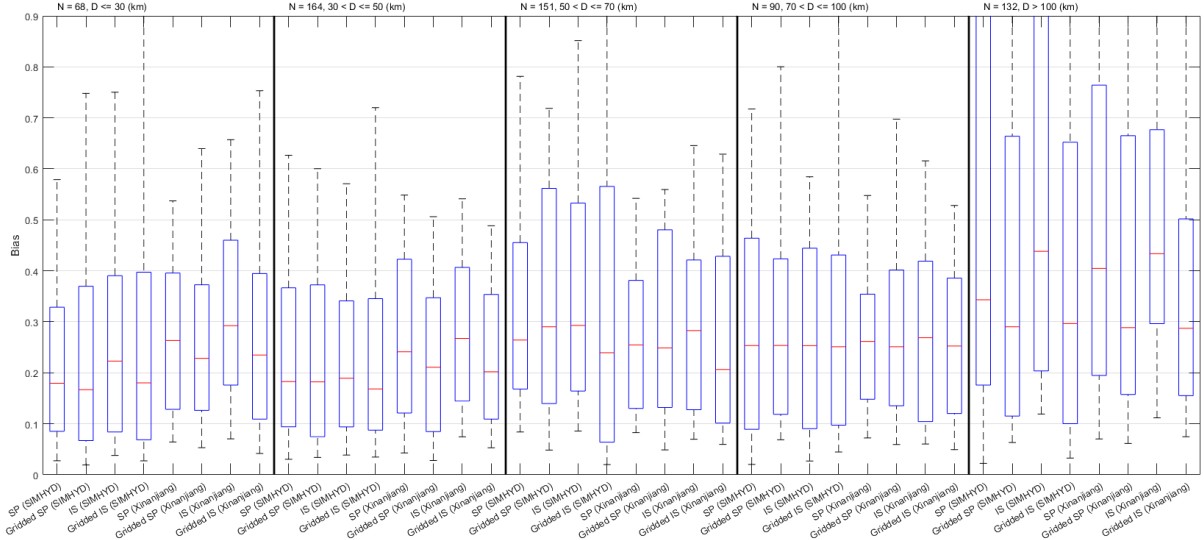

5    **Figure 10**. Stratification of bias regionalisation results using distance (km)





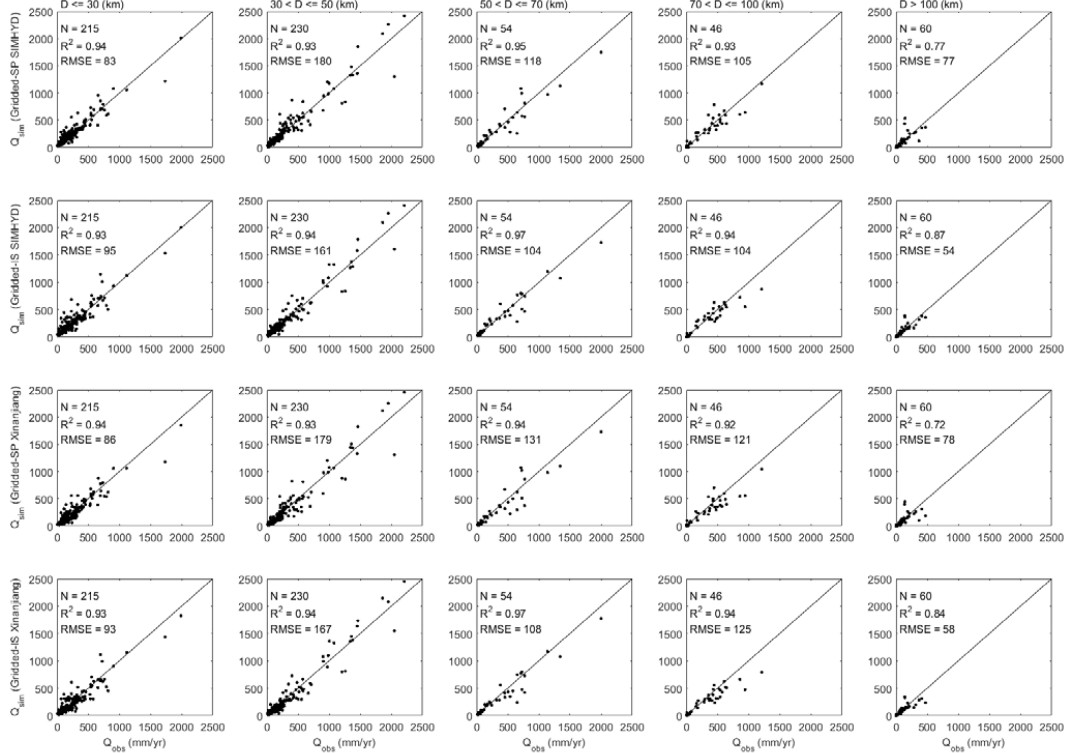

**Figure 11**. Comparison of the mean annual streamflow (mm/yr) between the observed ($Q_{obs}$) and the simulated

($Q_{sim}$) for the five groups of catchments stratified using distance. N is the sample number, $R^2$ is the coefficient

of determination, and RMSE is the root mean square error.

## 5. Discussion

### 5.1. The gridded IS approach is best for sparse and dry regions

Among the four regionalisation approaches, the gridded IS approach is best in the group(s) with greater

regionalisation distance or a dryer climate. There are 60 catchments belonging to the largest regionalisation

10    distance group (i.e., D > 100 km), and of them, there are 32 with $P < 600$ mm/yr and 13 with $600 \leq P < 800$

mm/yr. This indicates that the sparsely located streamflow gauges in Australia are mainly located in the semi-

arid or arid inland region, where the gridded IS approach is clearly superior to the other three. This is very important for improving the runoff prediction for continental Australia, the driest inhabited continent, and possibly for other large regions or continents.

## 5.2. The two rainfall–runoff models show consistent results

The SIMHYD model shows consistent modelling results for the Xinanjiang model, i.e., the relative merits among the four regionalisation approaches are the same. Both models indicate that the gridded IS approach is best in the dry and sparsely–located catchments and show the unnatural tessellated effect from the gridded SP approach. This indicates the common rules among these four regionalisation approaches in predicting runoff at a continental scale. The consistent model performance is similar to the multiple modelling results obtained

from southeastern Australia (Zhang and Chiew, 2009), France (Oudin et al. 2008), the Tibetan Plateau (Li, et al., 2014), and global catchments (Zhang et al., 2016).

## 5.3. Quality forcing data are the key for macro-scale runoff prediction

This study uses the best possible forcing data. All of the forcing data, including P, $T_{max}$, $T_{min}$, and radiation, were obtained from the approximately 5 km resolution dataset (Jeffrey et al., 2001). This guarantees that it

provides the best possible predictions in the runoff time series on continental scales. We further compared the runoff prediction results obtained from this study to other global/regional regionalisation studies. The median NSE obtained from this study is approximately 0.50 higher than that in Zhang et al. (2016), which used 0.5° resolution Princeton global forcing data and is approximately 0.90 higher than that obtained from the ensemble mean of eight global hydrological models in Beck et al. (2016b), which used 0.5° resolution WATCH global

forcing data ERA-interim (WFDEI) meteorological data. However, the median NSE obtained from this study is similar to or marginally different from that obtained in the regionalisation studies conducted in southeastern Australia (Vaze et al., 2011;Viney et al., 2009a), which all used the same SILO forcing data. Therefore, the key for improving the prediction of runoff at regional and continental scales is the quality of the forcing data.

### 5.4. Rainfall–runoff modelling together with the gridded IS approach should be widely used for macro-scale runoff prediction

This study demonstrates the ability of rainfall–runoff modelling together with the gridded IS approach for predicting the runoff time series in each $0.05° \times 0.05°$ resolution grid for continental Australia. The gridded

IS approach is superior to the gridded SP approach in the follow two aspects: it avoids the unnatural tessellated effect (Viney et al., 2014) and performs noticeably better in data sparse regions (Figures 9–11). Zhang et al. (2016) demonstrate that the SIMHYD and GR4J rainfall–runoff models, which are regionalised using the SP approach, are clearly superior to two land surface and global hydrological models for predicting runoff for 644 globally spread catchments. Beck et al. (2016b) also showed that the HBV rainfall–runoff model, which

was regionalised using the IS approach, outperformed nine land surface and global hydrological models for 1787 globally distributed catchments.

### 5.5. Practical ways for selecting predictors to build the gridded IS approach

It is necessary to select the IS predictors that are easily available and representative for macro-scale runoff prediction studies. This study chooses five predictors to build the similarity indices. Among them, the aridity

index reflects climate wetness or dryness; the fraction of forest ratio reflects the vegetation condition; the mean annual air temperature represents both climate and elevation; and the two rainfall indices represent rainfall seasonality and the standard deviation of daily rainfall. These predictors are relatively easily obtained and representative and are believed to be sufficient for continental Australia or other warm regions. It is possible that the current selected predictors are not enough for the high latitude northern hemisphere or high

elevation regions where snow melt is often a major contributor to runoff, and therefore, extra predictors, such as permanent snow cover, snowfall percentage, and days with a mean daily temperature less than 0°C, should be included as well.

### 6.    Conclusions



This study investigates four regionalisation methods (two obtained from spatial proximity and two obtained from integrated similarity) and two rainfall–runoff models (SIMHYD and Xinanjiang) to predict the daily runoff time series for 605 unregulated catchments that are widely distributed across continental Australia. The two models show consistent results that the margin among the four regionalisation approaches is very small

in wet and densely–located catchments. However, the gridded integrated similarity method outperformed the other three in the dry and sparsely–located catchments and overcomes the unnatural tessellated effect obtained from the gridded spatial proximity approach. Therefore, the gridded integrated similarity approach together with rainfall–runoff modelling is recommended for predicting daily runoff in each $0.05° \times 0.05°$ grid cell across continental Australia and other parts of the world. Additional studies should be focused on selecting

suitable predictors to build similarity indices.

**Acknowledgements**

This study is supported by the CSIRO Water for a Healthy Country Flagship runoff estimation strategic project (support No. R-02727-01). We would like to thank two anonymous reviewers and the associate editor for their thoughtful comments and suggestions.

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
