# Peer review of "Regionalising rainfall-runoff modelling for predicting daily runoff in continental Australia"

_Hydrology and Earth System Sciences, 2016_

## Referee Comment (RC1) · Anonymous Referee #1 · 16 Sep 2016

In this study four regionalisation methods are used to predict streamflow in continental Australia. The authors claim that regionalisation studies at the continental scale are almost non-existent in the literature. However, I do not believe that applying well known regionalisation methods to a different dataset represents a sufficiently novel contribution to the already extensive literature on this topic. Furthermore, it should not come as a surprise to the authors that spatial proximity methods do not provide the best results, when, for many ungauged catchments, the nearest gauges are located thousands of km away. Spatial proximity methods are not supposed to be used in such circumstances, and, therefore, I do not see the value in comparing this method with others in the context of the authors chosen case study.

In addition to these significant limitations of the paper, I have four other major concerns with this paper:

[Figure]

1) The results do not support the conclusions. I will provide some examples, but more can be found in the text. On page 11, lines 10-17, the authors report the median NSEsqrt and the median NSE for each regionalisation method for both SIMHYD and Xinanjiang model, which come from the boxplots shown in Figures 2 and 3. The authors state afterwards that 'This result suggests that the gridded SP approach is better than the SP approach ...' It is unclear to me whether the authors make such a statement based on the median value given in the previous sentences of that paragraph. If that is the case, this statement is incorrect as the median NSEsqrt for the SIMHYD model is higher for the SP approach (0.67) than for the gridded SP approach (0.66). The authors also state that 'the gridded IS approach slightly outperforms the IS approach' (page 10, lines 16-17), but the median NSEsqrt for the SIMHYD model for the IS approach (0.67) is the same as for the gridded IS approach (0.67). Other examples can be found on page 15, lines 12-15 and lines 16-18. On lines 12-16 the text reads 'For both the SIMHYD and Xinanjiang models, the IS and gridded IS approaches, respectively, outperform the SP and gridded SP approaches in the two groups with a large regionalisation distance ($70 < D \leq 100$ km and $D > 100$ km) but are marginally different from the SP and gridded approaches in the other three groups.' Looking at Figure 9, for $70 < D \leq 100$ km, I cannot conclude that IS (Xinanjiang) outperforms SP (Xinanjiang) or, for $D > 100$ km, that IS (SMYHD) outperforms SP (SMYHD). Furthermore, on what grounds do the authors assess whether the results of one method outperform another? This is not explained in the text, and significantly weakens the authors' statements about the validity and importance of their results. Statistical tests should be performed, to guarantee that such assertions are statistically significant.

2) The paper is poorly written, in particular section 2 onwards. Besides problems with grammar, there are many sentences that are challenging to understand and interpret. I will provide two examples, but more can be found in the text: 1) '(...) where NSEsqrt is Nash-Sutcliffe Efficiency of the daily square-root-transformed runoff data, which compromises weight when simulating high and low flow (...)' (page 8, lines 3-4); 2) 'For both the SIMHYD and Xinanjiang models, the IS and gridded IS approaches, respectively, outperform the SP and gridded SP approaches in the two dry catchment groups (P< 600 mm/yr and 600 $\leq$ P < 800 mm/yr), but are marginally different from the SP and gridded approaches in the wet catchment groups with a P > 800 mm/yr.' (page 15, lines 4-7). Some ideas are also expressed in a non-scientific manner, with little precise meaning or detail provided to the reader. As an example, '. . . the median NSE obtained from this study is similar to or marginally different from . . ..' (page 19, lines 20-21). A result cannot be both similar to, and marginally different. The paper does not appear to have been proof read, as the same information is repeated unnecessarily throughout the text. For example, at the beginning of section 2 (page 5, lines 5-6) it reads 'daily meteorological time series . . . from 1975 to 2012'. In the paragraph immediately after (page 5, line 21) it reads 'Data from 1975 to 2012 are used in this study.' Moreover, in section 2, page 5, lines 14-15, it reads 'The 0.05° × 0.05° SILO spatial data were averaged across all of the grid cells within a catchment to produce a catchment aver-age time series for use in this study.' and on page 6, lines 9-10, it reads 'The gridded data in each catchment were then extracted and averaged to obtain an aggregate daily data series for use in the modelling.'. Another curious example of the lack of attention to detail is that, in the acknowledgments, the authors thank two anonymous reviewers and the associate editor for their thoughtful comments and suggestions before the re-view process has even taken place. Some sections could also be better structured. For example, section 3.6 is comprised of only a single sentence.

3) Some of the choices made in the study are not justified in an adequate way. An example relates to the five properties used to define catchment similarity. Why were these five chosen and not others? Did the authors select these five properties based on statistical tests, literature, etc.? Similarly, the explanation for the use of five donor catchments (page 8, lines 18-22) is difficult to understand and needs to be more clearly explained so that the reader can judge the methods employed. Furthermore, why did the authors chose NSE, NSEsqrt and bias (section 3.5)? Finally, on page 10, line 2: the authors state 'This study used a value of 2 for p (Zhang et al., 2014b).' Why was a value of 2 used?

[Figure]

4) Some of the equations do not seem rigorous. For example, in Eq. (3) the authors use daily values, but in the explanation (page 8, lines 12-14) they mention mean annual runoff. Another example, refers to Eq. (7). The authors say that 'wk is the proportion of the grid cell within the "ungauged" catchment' (page 10, lines 9-10). If that is the case, why does the sum of wk from the L grid cells need to be 1, as the authors state on page 10, line 10? Lastly, the authors refer to Root Mean Square Error in the results section (page 15, line 21), but they have not defined it anywhere.

Given the many limitations in the paper, I cannot recommend publication of the manuscript in its current form and suggest that the paper should therefore be rejected. I hope that the authors will find these comments, while critical, to be useful in revising their manuscript for a future submission. Please note, however, that the list of examples given in this review is not exhaustive.

---

## Referee Comment (RC2) · Anonymous Referee #2 · 10 Nov 2016

**I. General comments**

The overall impression of this paper is that: not good structured, not clear on the purpose and novelty of this study, no sufficient support for the conclusion and poor language (not interesting to read and the text very hard to follow).

**II. Specific comments**

Page 1, abstract needs to be restructured: recapitulating the intention of the study, the novelty of the analysis and how it could be useful; key points about how these could be supported by the main findings. Page 2, introduction needs to be fulfilled with deep thinking on status quo, and what this study will bring or add on; with more insightful discussions on literature research. Page 4, line 1 and 2, why these 3 examples are listed here? Any particular reasons to select these from the long list in Table 1? Add

more discussions. Page 4, line 5, "may produce different conclusions between studies", here needs more details. Page 4, line 8, what "descriptors"? please elaborate. Page 4, line 10, not enough support to come to this conclusion. Page 4, line 12, it seems not true, there are many other studies, e.g. Oudin et al. 2008 Page 4, line 15-28, not sufficient argument why the authors chose these four methods, two models, and what's the value to compare the methods, models, and why it's applied to those catchments in Australia? Actually after reading the whole paper, still no clear idea on what's the purpose of this study and what's the benefit?

Page 5 and 6, Data section needs more details and to be addressed in a more meaningful way and in a logic structure. Just list a few example here, more can be found in text and need to be revised. Page 5, line 17, what's the meaning of "revised" rainfall-runoff models? Page 5, line 20, please add more details for the daily data of 605 catchments "collated by zhang et al. 2013" Page 5, line 21, please state why "data from 1975 to 2012 are used in this study". Page 5, line 26, & page6, any reference or source for "MODIS", "International Geosphere-Biosphere Programme"? There are many other similar things need to quote reference properly. Page 6, line 1-10, it's not clearly stated where and how data was obtained, produced, or processed. Please either cite original data sources, or data processing method, or quote reference properly, and in a meaningful, easy-understandable way.

Section 3-6, Poor language, poor structure, lack of detailed description, lack of meaningful discussion, no adequate justification through all these sections. Considerable modification will be required, and suggest having someone review the article before submission.

Why these objective functions were selected? How to do the model calibration and evaluation? What are the conditions to relate donor and ungauged catchments? Why select these five properties to define catchment similarity? Please embed more discussions and justifications in these sections, to lead to meaningful conclusion.

III. Technical comments

All the equations should be in a consistent format, and also for the paragraph after the equation which explains all the parameters in equation.

Many sections are too short to be a section, e.g. 3.5, 3.6, some has just one sentence.

In the 1st paragraph of section 4, all figures are mentioned together, this is not a good way to state the results. Please revise and prefer to talk about them one by one, with discussion.

Given the comments above, I won't recommend publication of this paper in current form, and suggest this paper will need major revisions for a future submission when applicable. I do hope these comments could be seen as constructive criticisms to help improve the overall paper and usefulness of the analysis.

---

## Author Comment (AC1) · 1 Dec 2016

First, we would like to thank the critical review from this anonymous referee, and thank the HESS editorial office to provide us an opportunity to clarify the concerns and address the queries. We have copied the comments one by one and each of them is followed by response (separated by 'End of this response' statement).

In this study four regionalisation methods are used to predict streamflow in continental Australia. The authors claim that regionalisation studies at the continental scale are almost non-existent in the literature. However, I do not believe that applying well known regionalisation methods to a different dataset represents a sufficiently novel contribution to the already extensive literature on this topic. Furthermore, it should not come as a surprise to the authors that spatial proximity methods do not provide the best

results, when, for many ungauged catchments, the nearest gauges are located thousands of km away. Spatial proximity methods are not supposed to be used in such circumstances, and, therefore, I do not see the value in comparing this method with others in the context of the authors chosen case study.

Re: It is indeed that there are few regionalisation studies carried out at the continental scale. We DID NOT claim that the regionalisation studies across a continental scale will be a sufficient novel contribution. Instead, we comprehensively compare the four regionalisation methods across the continental Australia. In addition, we feel a bit surprise that the merit of this study using 600+ catchments has not been pointed at all.

We DO NOT agree that there are no merits to apply spatial proximity method for far regional distance. There are no reports in literature that how the spatial proximity performs with the increase in regionalisation distance for Australian catchments. This is particularly important since most Australian catchments locate along coastal regions, but the inland and central Australia has very sparse gauges. It is not clear how the very uneven distribution of the catchments influence performance of different regionalisation approaches. Our study indeed demonstrates that use of the gridded integrated similarity approach outperforms the spatial proximity in data sparse inland and central Australia. (End of this response)

In addition to these significant limitations of the paper, I have four other major concerns with this paper: 1) The results do not support the conclusions. I will provide some examples, but more can be found in the text. On page 11, lines 10-17, the authors report the median NSEsqrt and the median NSE for each regionalisation method for both SIMHYD and Xinanjiang model, which come from the boxplots shown in Figures 2 and 3. The authors state afterwards that 'This result suggests that the gridded SP approach is better than the SP approach : : :' It is unclear to me whether the authors make such a statement based on the median value given in the previous sentences of that paragraph. If that is the case, this statement is incorrect as the median NSEsqrt for the SIMHYD model is higher for the SP approach (0.67) than for the gridded SP

approach (0.66).The authors also state that 'the gridded IS approach slightly outperforms the IS approach' (page 10, lines 16-17), but the median NSEsqrt for the SIMHYD model for the IS approach (0.67) is the same as for the gridded IS approach (0.67).

Re: Thanks for picking up some not accurate statements. But we do think that the results do not support the overall conclusions. It is indeed that that the four regionalisation methods show marginal difference in predicting the daily runoff in terms of NSEsqrt. We should clarify what you state is for NSE results. The median NSE for the SIMHYD model for the four methods (SP, gridded SP, IS, and gridded IS) is 0.55, 0.56, 0.54, and 0.58, respectively; the median NSE for the Xinanjiang model for the four methods is 0.54, 0.57, 0.54, and 0.57. This result suggests that in terms of NSE of daily runoff the gridded SP approach is better than the SP approach, and the gridded IS approach slightly outperforms the IS approach. (End of this response)

Other examples can be found on page 15, lines 12-15 and lines 16-18. On lines 12-16 the text reads 'For both the SIMHYD and Xinanjiang models, the IS and gridded IS approaches, respectively, outperform the SP and gridded SP approaches in the two groups with a large regionalisation distance (70 < D <= 100 km and D > 100 km) but are marginally different from the SP and gridded approaches in the other three groups.' Looking at Figure 9, for 70 < D <= 100 km, I cannot conclude that IS (Xinanjiang) outperforms SP (Xinanjiang) or, for D > 100 km, that IS (SMYHD) outperforms SP (SMYHD). Furthermore, on what grounds do the authors assess whether the results of one method outperform another? This is not explained in the text, and significantly weakens the authors' statements about the validity and importance of their results. Statistical tests should be performed, to guarantee that such assertions are statistically significant.

Re: We should clarify that one approach outperforms another based on the NSE difference more than 0.02 and the two approaches perform similarly when the difference is smaller than 0.02 (Zhang and Chiew, 2009). We should state the case of Xinanjiang with 70 < D <= 100 km as follows " these two approaches are similarly for Xinanjiang

model with 70 < D <= 100 km. (End of this response)

2) The paper is poorly written, in particular section 2 onwards. Besides problems with grammar, there are many sentences that are challenging to understand and interpret. I will provide two examples, but more can be found in the text: 1) '(: : :) where NSEsqrt is Nash-Sutcliffe Efficiency of the daily square-root-transformed runoff data, which compromises weight when simulating high and low flow (: : :)' (page 8, lines 3-4); 2) 'For both the SIMHYD and Xinanjiang models, the IS and gridded IS approaches, respectively, outperform the SP and gridded SP approaches in the two dry catchment groups (P< 600 mm/yr and 600 _ P < 800 mm/yr), but are marginally different from the SP and gridded approaches in the wet catchment groups with a P > 800 mm/yr.' (page 15, lines 4-7). Some ideas are also expressed in a non-scientific manner, with little precise meaning or detail provided to the reader. As an example, ': : : the median NSE obtained from this study is similar to or marginally different from : : :.' (page 19, lines 20-21). A result cannot be both similar to, and marginally different. The paper does not appear to have been proof read, as the same information is repeated unnecessarily throughout the text. For example, at the beginning of section 2 (page 5, lines 5-6) it reads 'daily meteorological time series : : : from 1975 to 2012'. In the paragraph immediately after (page 5, line 21) it reads 'Data from 1975 to 2012 are used in this study.' Moreover, in section 2, page 5, lines 14-15, it reads 'The 0.05_ _ 0.05_ SILO spatial data were averaged across all of the grid cells within a catchment to produce a catchment average time series for use in this study.' and on page 6, lines 9-10, it reads 'The gridded data in each catchment were then extracted and averaged to obtain an aggregate daily data series for use in the modelling.'. Another curious example of the lack of attention to detail is that, in the acknowledgments, the authors thank two anonymous reviewers and the associate editor for their thoughtful comments and suggestions before the review process has even taken place. Some sections could also be better structured. For example, section 3.6 is comprised of only a single sentence.

Re: Thanks for the reviewer for picking up grammar issues. Sorry for the careless mistakes to confuse this reviewer. Having said that, this manuscript has been proof-read by a professor editorial company – the American Journal Expert. I feel very disappointed that the reviewer still pick up the grammar issue. We will do a thorough grammar correction.

In terms of the acknowledgement, this reviewer's thoughtful thinking will make us to present our manuscript more accurately. We will follow his/her suggestions for rephrase text and manuscript structure.

(End of this response)

3) Some of the choices made in the study are not justified in an adequate way. An example relates to the five properties used to define catchment similarity. Why were these five chosen and not others? Did the authors select these five properties based on statistical tests, literature, etc.? Similarly, the explanation for the use of five donor catchments (page 8, lines 18-22) is difficult to understand and needs to be more clearly explained so that the reader can judge the methods employed. Furthermore, why did the authors chose NSE, NSEsqrt and bias (section 3.5)? Finally, on page 10, line 2: the authors state 'This study used a value of 2 for p (Zhang et al., 2014b).' Why was a value of 2 used?

Re: in terms of choice of the five properties, we have had a section to discuss this. In section 5.5, the text states that "5.5 Practical ways for selecting predictors to build the gridded IS approach. It is necessary to select the IS predictors that are easily available and representative for macro-scale runoff prediction studies. This study chooses five predictors to build the similarity indices. Among them, the aridity index reflects climate wetness or dryness; the fraction of forest ratio reflects the vegetation condition; the mean annual air temperature represents both climate and elevation; and the two rainfall indices represent rainfall seasonality and the standard deviation of daily rainfall. These predictors are relatively easily obtained and representative and are believed to be sufficient for continental Australia or other warm regions. It is possible that the

current selected predictors are not enough for the high latitude northern hemisphere or high elevation regions where snow melt is often a major contributor to runoff, and therefore, extra predictors, such as permanent snow cover, snowfall percentage, and days with a mean daily temperature less than 0°C, should be included as well.". We should put more argument for the choice, such as we did correlation analysis first and picked up the five with good correlations and they are representative.

In terms of use of five donor catchments, we should more clearly explain that. We did this based on numerous donor catchment number sensitivity analysis, as indicated by Zhang and Chiew, (2009) and Oudin et al. (2008)

In terms of chose of NSE, NSEsqrt and bias, these metrics are standard ones used for evaluating runoff estimates from rainfall-runoff modelling. The NSE focus on high daily flow evaluation; the NSEsqrt focuses on not only high daily flow but low daily flow as well; the bias is the evaluation of accuracy for mean annual runoff.

In terms of 2 for p, this is the default parameter used in the inverse distance weight approach. Ideally, we can optimise this parameter and do the sensitivity analysis on the weight parameter by varying it within a certain range. However, it is out of scope of this study.

(End of this response)

4) Some of the equations do not seem rigorous. For example, in Eq. (3) the authors use daily values, but in the explanation (page 8, lines 12-14) they mention mean annual runoff. Another example, refers to Eq. (7). The authors say that 'wk is the proportion of the grid cell within the "ungauged" catchment' (page 10, lines 9-10). If that is the case, why does the sum of wk from the L grid cells need to be 1, as the authors state on page 10, line 10? Lastly, the authors refer to Root Mean Square Error in the results section (page 15, line 21), but they have not defined it anywhere.

Given the many limitations in the paper, I cannot recommend publication of the

manuscript in its current form and suggest that the paper should therefore be rejected. I hope that the authors will find these comments, while critical, to be useful in revising their manuscript for a future submission. Please note, however, that the list of examples given in this review is not exhaustive.

Re: Eq. (3). Yes, we should mention mean daily runoff.

'wk' issue. This is bad English. When we talk about the proportion, the sum of wk is 100; when we talk about the ratio, the sum of wk is 1.

'rmse'. Yes, we will define rmse before we use it.

In summary, we thank dearly to this reviewer for his/her thoughtful thinking and detailed points which can easily help us to improve our MS during next round of revision. Overall the critical points do not influence the key conclusions drawn from this study. The merit of this study should be more clearly communicated and articulated. We are preparing to do so. (End of this response)

---

## Author Comment (AC2) · 6 Dec 2016

First, we would like to thank the critical review from this anonymous referee, and thank the HESS editorial office to provide us an opportunity to clarify the concerns and address the queries. We have copied the comments one by one and each of them is followed by response (Re:).

I. General comments The overall impression of this paper is that: not good structured, not clear on the purpose and novelty of this study, no sufficient support for the conclusion and poor language (not interesting to read and the text very hard to follow)..

Re: This comment is not professional at all. It seems that it can apply to any manuscript. It is indeed astonished. We believe that this paper has enough novelty deserving to be published in HESS. The novelty of this study includes that (1)

[Figure]

it use 600+ catchments for continental regionalisation study, which is indeed one of few studies focusing on such a large scale; (2) it comprehensively compares the four regionalisation methods across the continental Australia; (3) it demonstrates that using the gridded integrated similarity approach outperforms using the spatial proximity in data sparse inland and central Australia, which is particularly important since most Australian catchments locate along coastal regions, but the inland and central Australia has very sparse gauges. We acknowledge the criticism that poor language was used. We will try our best to improve its readability in the next version if the editor provides us a revision opportunity. (End of response)

II. Specific comments Page 1, abstract needs to be restructured: recapitulating the intention of the study, the novelty of the analysis and how it could be useful; key points about how these could be supported by the main findings. Page 2, introduction needs to be fulfilled with deep thinking on status quo, and what this study will bring or add on; with more insightful discussions on literature research. Page 4, line 1 and 2, why these 3 examples are listed here? Any particular reasons to select these from the long list in Table 1? Add. more discussions. Page 4, line 5, "may produce different conclusions between studies", here needs more details. Page 4, line 8, what "descriptors"? please elaborate. Page 4, line 10, not enough support to come to this conclusion. Page 4, line 12, it seems not true, there are many other studies, e.g. Oudin et al. 2008 Page 4, line 15-28, not sufficient argument why the authors chose these four methods, two models, and what's the value to compare the methods, models, and why it's applied to those catchments in Australia? Actually after reading the whole paper, still no clear idea on what's the purpose of this study and what's the benefit?

Re: Page 1, yes we would like to restructure abstract to articulate the intention and novelty of this study, and then show the key results followed by main findings. Page 2, yes we would like to have deeper thinking on status quo. Page 4, we need to clarify that these are just examples and more examples are summarised in Table 1. We will have more discussion on Table 1. We would like to show more details on

the statement "may produce different conclusion between studies". Page 4, line 8, "descriptors" should be replaced by "characteristics". Page 4, line 10, we would soften our language for doing large-scale regionalisation studies Page 4, line 12, Oudin et al. 2008 only used 913 France catchments, noting to do with continental daily runoff prediction study. The reviewer states "it seems that it is not true...", he or she DID NOT provide any reference to prove what our statement is not correct. Page 4, line 15-28, it is really astonished to see this kind of comment ".. chose four methods, two models...". Choose of two models can make our conclusion more robust. Are there any wrong with this? Comparing four regionalisation methods can have a comprehensive evaluation of the various regionalisation methods. Are there any wrong with that? We will try to articulate the manuscript objectives. (End of response)

Page 5 and 6, Data section needs more details and to be addressed in a more meaningful way and in a logic structure. Just list a few example here, more can be found in text and need to be revised. Page 5, line 17, what's the meaning of "revised" rainfallrunoff models? Page 5, line 20, please add more details for the daily data of 605 catchments "collated by zhang et al. 2013" Page 5, line 21, please state why "data from 1975 to 2012 are used in this study". Page 5, line 26, & page6, any reference or source for "MODIS", "International Geosphere-Biosphere Programme"? There are many other similar things need to quote reference properly. Page 6, line 1-10, it's not clearly stated where and how data was obtained, produced, or processed. Please either cite original data sources, or data processing method, or quote reference properly, and in a meaningful, easy-understandable way..

Re:. Page 5, line 17, what's the meaning of "revised" rainfallrunoff models? We should clarify here that the detail for the revised rainfall-runoff model is introduced in section 3.1 Page 5, line 20, please add more details for the daily data of 605 catchments. Yes, we would put more details on the daily runoff data of 605 catchments. why "data from 1975 to 2012 are used in this study". This data set comes from the national water accounting project which collocated daily streamflow data for 1975 to 2012, which

cover long-period of time, and different climate conditions, being good enough for any regionalisation studies.

Modis issue. We will show a reference for the MODIS dataset. Yes, we will show more details on IGBP land cover types

We will introduce more clearly on data source, data processing processing method and quote reference appropriately in next round of revision. (End of response)

Section 3-6, Poor language, poor structure, lack of detailed description, lack of meaningful discussion, no adequate justification through all these sections. Considerable modification will be required, and suggest having someone review the article before submission.

Re: We accept the criticism of this review. We will try our best to more accurately summarise our results, to have more meaningful and thoughtful discussion, and will ask a peer to review it again

This reviewer's thoughtful thinking will make us to present our manuscript more accurately. We will follow his/her suggestions for rephrase text and manuscript structure. . (End of response)

Why these objective functions were selected? How to do the model calibration and evaluation? What are the conditions to relate donor and ungauged catchments? Why select these five properties to define catchment similarity? Please embed more discussions and justifications in these sections, to lead to meaningful conclusion.

Re: The objective function using NSEsqrt focuses on not only high daily flow but low daily flow as well; the bias is the evaluation of accuracy for mean annual runoff. This is the objective function widely used for rainfall-runoff modelling.

Model calibration. A global model calibration method, the genetic algorithm, was used for model calibration. Model evaluation. Regionalisation evaluation. We should clarify this, should not we? Conditions of donor and ungauged catchments. we should clarify

that all the catchments selected are met several criteria: (1) unregulated; (2) without subject to noticeable urbanisation; (3) streamflow data length more than 10 years; (4) catchment area less than 5000 km2.

The five properties. We have had a section to discuss this. In section 5.5, the text states that "5.5 Practical ways for selecting predictors to build the gridded IS approach. It is necessary to select the IS predictors that are easily available and representative for macro-scale runoff prediction studies. This study chooses five predictors to build the similarity indices. Among them, the aridity index reflects climate wetness or dryness; the fraction of forest ratio reflects the vegetation condition; the mean annual air temperature represents both climate and elevation; and the two rainfall indices represent rainfall seasonality and the standard deviation of daily rainfall. These predictors are relatively easily obtained and representative and are believed to be sufficient for continental Australia or other warm regions. It is possible that the current selected predictors are not enough for the high latitude northern hemisphere or high elevation regions where snow melt is often a major contributor to runoff, and therefore, extra predictors, such as permanent snow cover, snowfall percentage, and days with a mean daily temperature less than $0°C$, should be included as well.". We should put more argument for the choice, such as we did correlation analysis first and picked up the five with good correlations and they are representative.

Five donor catchments, we should more clearly explain that. We did this based on numerous donor catchment number sensitivity analysis, as indicated by Zhang and Chiew, (2009) and Oudin et al. (2008). (End of response)

III. Technical comments All the equations should be in a consistent format, and also for the paragraph after the equation which explains all the parameters in equation. Many sections are too short to be a section, e.g. 3.5, 3.6, some has just one sentence. In the 1st paragraph of section 4, all figures are mentioned together, this is not a good way to state the results. Please revise and prefer to talk about them one by one, with discussion.

Re: We accept the criticism, and are preparing to improve the manuscript as suggested by this reviewer.

In summary, we thank dearly to this reviewer for his/her thoughtful thinking and detailed points which can easily help us to improve our MS during next round of revision. Overall the critical points do not influence the key conclusions drawn from this study. The merit of this study should be more clearly communicated and articulated. We are preparing to do so. (End of response)

―――――――――――――――――